# *Krüppel-homologue 1* Mediates Hormonally Regulated Dominance Rank in a Social Bee

**DOI:** 10.3390/biology10111188

**Published:** 2021-11-15

**Authors:** Atul Pandey, Guy Bloch

**Affiliations:** 1Department of Ecology, Evolution and Behavior, The Hebrew University of Jerusalem, Jerusalem 9190401, Israel; 2Department of Ecology and Evolutionary Biology, University of Michigan, Ann Arbor, MI 48109, USA

**Keywords:** *Krüppel-homologue 1*, juvenile hormone, vitellogenin, RNA interference, nanoparticles, reproduction, dominance, aggression, oogenesis

## Abstract

**Simple Summary:**

In diverse animal species, individuals establish dominance hierarchies by means of agonistic interactions. Dominance rank is functionally significant because it affects access to vital resources such as mates, food, and shelter, but little is known on the underlying genetic and molecular mechanisms, specifically in insects, and among females. We tested the hypothesis that *Krüppel homologue 1*, a key endocrine signaling gene, influences dominance among bumble bee female workers. We developed and validated a new nanoparticle-based protocol to down-regulate gene expression in bumble bees. Using this protocol, we show that *Krüppel homologue 1* mediates endocrine influences, not only on fertility and wax production, but also on aggression and dominance rank. These findings, which establish the first causal link between a gene and a dominance rank in a social insect, are important for determining whether there are general molecular principles governing dominance ranks across gender and animal species.

**Abstract:**

Dominance hierarchies are ubiquitous in invertebrates and vertebrates, but little is known on how genes influence dominance rank. Our gaps in knowledge are specifically significant concerning female hierarchies, particularly in insects. To start filling these gaps, we studied the social bumble bee *Bombus terrestris*, in which social hierarchies among females are common and functionally significant. Dominance rank in this bee is influenced by multiple factors, including juvenile hormone (JH) that is a major gonadotropin in this species. We tested the hypothesis that the JH responsive transcription factor *Krüppel homologue 1* (*Kr-h1*) mediates hormonal influences on dominance behavior. We first developed and validated a perfluorocarbon nanoparticles-based RNA interference protocol for knocking down *Kr-h1* expression. We then used this procedure to show that *Kr-h1* mediates the influence of JH, not only on oogenesis and wax production, but also on aggression and dominance rank. To the best of our knowledge, this is the first study causally linking a gene to dominance rank in social insects, and one of only a few such studies on insects or on female hierarchies. These findings are important for determining whether there are general molecular principles governing dominance rank across gender and taxa.

## 1. Introduction

Social hierarchies are ubiquitous in invertebrates and vertebrates. Behavioral dominance is the basic principle organizing social hierarchies and is typically established by the outcome of multiple contests between pairs of conspecifics which are resolved with a clear dominance rank. Dominance rank is functionally significant given that high-ranked individuals typically gain priority access to resources, shelters, and mating opportunities [1,2,3,4]. Low dominance ranks, on the other hand, are commonly associated with stress and compromised physical conditions in many animals, including humans, even in systems with no apparent rank-related asymmetries with access to resources [5,6,7,8].

Although it is well established that dominance rank has fitness consequences, the possibility that social dominance can be genetically inherited has been a subject of recurrent debate [9,10]. The difficulties in linking dominance to specific genes stem from evidence that social rank is influenced by multiple factors that may interact in complex manners. There is evidence that a dominance rank can be passed on through social (e.g., matriline or network of allies) rather than genetic mechanisms [9]. An additional difficulty is that dominance rank is inherently context-dependent and may be expressed in only limited situations or with specific partners. The same individual may be dominant over one partner and submissive with another [11]. The outcome of a dominance hierarchy contest is commonly influenced by the individual intrinsic Recourse Holding Potential (RHP), which relates to attributes such as body size, mass, physical strength, resting metabolism, and fighting ability [12]. In addition, prior experience (e.g., winner, loser, or bystander effects; chronological age; fighting experience), and the subjective evaluation of a resource, including motivation, may also be crucial, adding additional levels of complexity [13,14]. Thus, it is not obvious how such group-level, context-dependent traits can be genetically inherited [10]. In spite of these difficulties, artificial selection in laboratory studies with various species including rodents [15,16] and insects [17], have shown that dominance rank can be selected, lending credence to the notion that social rank is influenced by genes. However, little is known of the identity of specific genes underlying the expression of dominance behavior and the mechanisms by which they influence dominance rank [9].

Genes related to dominance rank have been studied mostly in mammals and fishes. In some of these model systems, there has been significant progress in describing and understanding the neural circuits that control social status, and in linking specific genes to these pathways [18,19]. These studies have repeatedly associated dominance rank with certain key pathways, and particularly serotonergic and dopaminergic systems [9], and genes related to oxytocin and vasopressin signaling [20,21]. Some studies have established causal relationships between a specific gene product and social dominance (for example, [22,23]). However, the interpretation of the results of gene manipulation or mutation studies can be challenging given that these monoaminergic and peptidergic systems influence many neuronal processes and may affect motivation, mood, or agonistic behavior, regardless of social rank [24]. For example, mice with knockout *serotonin transporter* (*SLC6A4*) show reduced social dominance, but also display increased anxiety and reduced locomotion, which can decrease their motivation to be involved in an aggressive contest [25,26]. An additional complexity is that the link between aggression and dominance is not always simple, such that initial aggression may not be a good predictor of later dominance [27,28,29].

Relative to mammals and other vertebrates, little is known on the molecular and neuronal underpinnings governing dominance in insects. Given that social hierarchy behavior in mammals and other vertebrates depends on a collection of intricate cognitive traits [30,31,32,33], it is not clear to what extent the neuronal and molecular processes underlying social dominance are similar in vertebrates and invertebrates. Perhaps the best studied dominance-related gene in invertebrates is *fruitless*, which regulates courtship behavior and also contributes to gender-specific differences in dominance-related aggression of the fruit fly *Drosophila melanogaster* [34,35]. Dominance interactions are also a hallmark of social insects such as *Polistes* paper wasps, bumble bees, and queenless ants, which live in relatively small and simple societies (commonly dubbed “primitively eusocial”, [36]). As in other species, dominance is functionally significant because top-ranked females are more likely to reproduce [37]. Despite the long tradition of research on dominance in social insects [36,38], little is still known about the genes and molecular processes influencing dominance in this important group of insects. In contrast to studies with solitary insects such as *Drosophila*, in which dominance has been studied mostly in males in the context of access to receptive females [39], in social insects, dominance hierarchies are typically formed between females. This includes interactions between queens and between groups of unmated workers laying unfertilized haploid eggs that develop into males [37]. Given evidence in mammals that the regulation of dominance differs between males and females [40,41], it is also not clear to what extent implicating particular genes in social dominance in male flies is relevant to understanding dominance in social insects. Nevertheless, dominance behavior or dominance rank were correlated with certain gene expression patterns in social hierarchies between queens in highly social species (e.g., paper wasps [42,43] fire ants [44]), as well as in subsocial and facultatively social bees [45,46], and also between social insect workers [47,48]. The expression of some of the same genes (e.g., *vg*, *beadex*) and molecular processes (e.g., the insulin signaling pathway) were found in studies on multiple species and social systems, which is consistent with a conserved role in the regulation of dominance rank [43,45,46,48].

A common theme in the regulation of dominance rank in vertebrate and insects is the crucial role of the endocrine system. Studies with many species of vertebrates have established that social dominance is tightly regulated by various endocrine signals, and specifically the stress and sex steroid hormones [19,40,49,50]. In social insects, the best studied endocrine signal in the context of dominance rank is juvenile hormone (JH), a pleotropic hormone known to regulate diverse developmental, physiological, and behavioral processes in insects [51,52,53,54,55,56,57]. JH functions as a major gonadotropic hormone regulating fertility and reproductive physiology in most species studied so far [51,58,59]. However, there seems to be significant interspecies variability in adult insects, with evidence for species in which JH does not function as a gonadotropin [60,61,62,63]. For example, whereas in solitary bees, JH seems to retain its gonadotropic functions, in the highly eusocial honey bee it influences age-related division of labor, but not reproduction [64,65]. 

In *Bombus terrestris*, the best studied bumble bee, JH functions as a major gonadotropin coordinating multiple tissues and processes related to reproduction [47,66,67,68,69,70]. This includes oogenesis, vitellogenesis, exocrine gland activity, wax deposition, egg-laying, metabolic rate, and the expression of hundreds of brain and fat body transcripts [47,70,71]. Given the close association between dominance and reproduction in bumble bees, it has been long suspected that JH influences dominance, as well as other behaviors related to reproduction [66,67,72,73]. This hypothesis has been recently supported in JH manipulation experiments in which JH titers were reduced by surgical or chemical corpora allata (CA) nullification, and subsequently elevated by replacement therapy with JH-III, the natural JH of bumble bees [69,74]. These manipulation studies show that the relationships between JH, dominance, and aggression, are not simple. An individual with reduced JH levels is less likely to acquire a high dominance rank in small groups of orphan (“queenless”) workers, and this effect is fully reverted by topical treatment with JH-III. However, a clear dominance hierarchy was also established in groups in which all individuals are similarly manipulated to have either high or low JH levels. In these groups, body size and previous experience were good predictors of dominance rank. Thus, JH appears to be one of several factors influencing dominance behavior in bumble bees [69].

Given the established role of reproductive hormones in the regulation of dominance in vertebrates (see above) and the evidence that JH influences dominance in bumble bees and other social insects, we hypothesized that JH signaling pathway genes may influence dominance-related behaviors. To start testing this hypothesis, we focused on the C2H2 zinc-finger transcription factor *Krupple-homolog 1* (*Kr-h1*), which is tightly regulated by JH in many tissues and is considered a pivotal JH signaling gene in insects [75,76,77,78,79,80,81]. The expression of *Kr-h1* is high in the JH target tissues such as the brain, ovaries, antennae, fat bodies and ovaries, that are associated with reproduction and behavior [80,81,82,83]. In insects from diverse orders (e.g., the honey bee *Apis mellifera* [84], the fruit fly, *Drosophila melanogaster* [85], Oriental fruit fly, *Bactrocera dorsalis* [77], Migratory locust, *Locusta migratoria* [86] and Red flour beetle, *Tribolium castaneum* [87]), *Kr-h1* functions as an early JH inducible gene acting downstream of the JH receptor [55,88,89,90] to regulate the expression of multiple pathways [89,91,92,93]. Studies in which *Kr-h1* levels were knocked down using RNA interference (RNAi) approaches lend additional support to the premise that *Kr-h1* mediates the influence of JH on oogenesis and fertility (e.g., *Locusta migratoria* [86]; *Bactrocera dorsalis* [77]; White-back planthopper, *Sogatella furcifera* [94]; Brown planthopper *Nilaparvata lugens* [95,96]; Asiatic rice borer, *Chilo suppressalis* [76]; Cotton Bollworm, *Helicoverpa armigera* [97]; and Red flour beetle *Tribolium castaneum* [98]). These studies show that *Kr-h1* is a key gene mediating the influence of JH on physiology and behavior, making it an excellent candidate gene for studies on the molecular basis of dominance behavior.

In bumble bee workers, brain *Kr-h1* transcript abundance is positively correlated with ovarian activity [83]. JH manipulation studies further show that *Kr-h1* expression in the brain and fat bodies is upregulated by JH [47,70,83]. Studies with bumble bee queens are, however, less consistent. Amsalem et al. (2015) reported a higher fat body *Kr-h1* levels in nest foundresses (which typically have high JH titers) compared to pre-diapause mated queens (with typically low JH levels) [99]. Jedlicka et al. (2016) reported higher levels in the fat body and other tissues (crop, hypopharyngeal glands, ventriculus, rectum) in diapausing (with typically low JH titers) compared to virgin gynes (low JH titers) and egg-laying queens (high JH titers) [100]. 

In order to test the hypothesis that *Kr-h1* mediates the influence of JH signaling on dominance, we developed and validated a new RNAi procedure for down-regulating *Kr-h1* RNA levels. We chose a gene knockdown approach rather than mutagenesis because JH signaling plays important roles during larval development and metamorphosis. We load the double strand RNA (dsRNA) onto nanoparticles allowing us to introduce relatively small amounts of RNA interfering with dsRNA and thus minimizing RNA toxicity and nonspecific effects. Using this improved protocol, we show that *Kr-h1* not only mediates the influence of JH on fertility and wax production, but also on dominance and aggression.

## 2. Materials and Methods

### 2.1. Bees

We purchased incipient *B. terrestris* colonies from Polyam Pollination Services, Kibbutz, Yad-Mordechai, Israel, or Bio-Bee Biological Systems Ltd., Kibbutz Sde-Eliyahu, Israel. The colonies arrived at approximately 2–4 days post first worker emergence, and contained a queen, 5–10 workers, and brood at various stages of development. We housed each colony in a wooden nesting box (21 × 21 × 12 cm) with top and front walls made of transparent Plexiglass panels, enabling a clear view of the entire wax comb. All the nest boxes were placed in an environmental chamber (29 ± 1 °C; 55% ± 10% RH, continuously monitored with Hobo UX100-003 data loggers) at the Bee Research Facility at the Edmond J. Safra Campus of the Hebrew University of Jerusalem, Givat Ram, Jerusalem, Israel. We provisioned the colonies with ad libitum commercial sugar syrup and fresh honey bee-collected pollen (obtained from Polyam Pollination Services, Kibbutz, Yad-Mordechai, Israel). For most of the experiments we used newly emerged (callow) workers. Callow bees (up to about 12 h of age) do not yet show yellow pigmentation and were identified based on their white coloration. For all the experiments detailed below, we used groups of four orphan (queenless) workers. Additional details are provided in the sections describing the specific experiments.

### 2.2. Behavioral Observations

We performed behavioral observations as detailed in our previous study [69]. The days in which we performed observations differed between the experiments as reported for each experiment in the sections below. Observation sessions lasted 20–30 min and the measurements are presented as events per 30 min. We recorded threatening displays that include “*buzzing*”, and “*pumping*” [69,101,102,103]. Buzzing displays are characterized by fast, short wing vibrations of a worker facing another worker bee; “*pumping*” is characterized by abdominal contraction/extension movements (“*pumping*”) performed by a bee standing and facing a nestmate bee. We also recorded overt aggression, which typically follows threatening displays, and includes darting and attacks directed towards another bee. To estimate the dominance rank of individual bees within groups, we calculated for each bee a Dominance Index (DI) as in previous studies [69,101,103,104]. Briefly, the DI is the sum proportion of encounters between each pair of bees in which the focal bee did not retreat out of the total encounters (1- (Retreats/Total encounters)). Thus, it includes both encounters with and without threatening displays, but in which it was clear that the bees assessed each other. The worker with the highest DI value was termed “*alpha*” (α), followed by “*beta*” (β), “*gamma*” (γ) and “*delta*” (δ) in descending DI order. The observer was blind to the treatment to which the bees were subjected. Bees were individually tagged with numbered plastic disks (Opalith tags, Graze, factory beekeeping equipment, Germany) or painted with xylene free silver color paint (Pilot-PL01735), using a distinct pattern for each bee in a group to allow individual identification. In order to minimize the observer effects and other destructions, we observed the bees under dim red light and minimized vibrations that could potentially startle the bees.

### 2.3. Assessing Ovarian Activity and Wax Deposition

At the end of each experiment, we stored the focal bees at −20 °C. To assess ovarian state, we fixed the bee on a wax-filled dissecting plate under a stereomicroscope (Nikon SMZ645) and immersed it in honey bee saline (Huang et al., 1991). We cut three incisions through the lateral and ventral abdominal cuticle using fine scissors and immersed the internal organs in saline. We gently removed the ovaries into a drop of saline on a microscope slide and used an ocular ruler to measure the length of the four largest terminal oocytes in the ovaries (*B. terrestris* females typically have 8 ovarioles). We used the average terminal oocyte length at seven days of age as an index for ovarian activity. At this age, queenless workers typically show the entire range of terminal oocyte size, including bees with mature eggs, but still do not show signs of reabsorption [101,105]. To estimate the amount of wax deposited in a cage, we collected all the wax in the cage (honeypots, egg cups, etc.) and removed nectar, pollen, and eggs deposited in the cups, as needed. We scraped all the wax attached to the cardboard bottom while trying to avoid any pollen remains and bee poops. We absorbed the whole wax collected with paper towel, removing any liquid or moisture from it. The total amount of wax collected from a cage was weighed using an electronic balance.

### 2.4. Reverse-Transcription Real-Time Quantitative PCR (qPCR)

We separated the frozen head from the rest of the body and lightly freeze-dried the heads for 150 min (−50 °C, Heto Drywinner, Thermo scientific, Waltham, MA, USA). The frozen dried brains were stored in −70 °C until further analyses. In order to extract RNA from the abdominal fat body, we immersed the whole abdomen in RNA save solution (Biological Industries, SKU-01-891-1B). We made two lateral ventral longitudinal incisions and an additional connecting frontal cut, allowing us to remove the cuticle and expose the inner abdominal organs. We then carefully removed the digestive system, Malpighian tubes, ovaries, and any other tissue besides the fat body connected to the abdominal cuticle. The fat body tissue was separated from the cuticle and was stored in RNA Save solution for molecular analyses. We extracted total RNA using the Bioline Isolate II RNA extraction Kit according to the manufacturer’s instructions and assessed RNA quantity and quality using Nanodrop and agarose gel electrophoresis (0.8%). Complementary DNA (cDNA) was reverse-transcribed using the same protocol described in Section 2.5. We used the cDNA as a template in a qPCR reaction on Applied Biosystems StepOne Real-Time PCR sequence analyzer using the fast SYBR green master mix (Applied Biosystems, Cat# 4385612). We included three technical replicates for each biological sample. We chose *Elongation factor 1α (Ef-1α)* as the housekeeping loading control gene, given that its expression was unaffected by JH manipulation in previous studies [47,106]. We performed all the statistical tests on ΔCt values, which are normally distributed. For the graphical display, we used the relative quantification (RQ) expression representation calculated from ΔΔCt values and represented relative to the control group (which received an average level = 1.0).

### 2.5. Synthesis and Delivery of dsRNA

We used the siRNA sequence generator web-tool (Gene script, https://www.genscript.com/tools/sirna-target-finder, accessed on 21 May 2016) to predict the possible siRNAs products of our *Kr-h1* and *pGEM-T* (control) dsRNA sequences, helping us to design effective and specific dsRNA constructs. These predicted siRNAs were further screened for possible off-target effects. To generate a template for synthesizing *Kr-h1* dsRNA constructs, we extracted total RNA from the fat bodies of individual bumble bees (Section 2.4) and reverse-transcribed it (300 ng/reaction) following the manufacturer’s protocol (Cat. Bio-27036, Bioline meridian biosciences, USA). The synthesized cDNA was then used as a template in polymerase chain reactions (PCR), with a first pair of primers designed to amplify a 668 bp long fragment (Appendix A, row 1). This DNA product was purified and cleaned using DNA clean and concentrator-5 kit (Cat. D4003, Zymo Research, Murphy Ave. Irvine, CA, USA) and used it as a template for a PCR with T7 promotor sequence linked to nested primers designed to amplify a 440 bp long *Kr-h1* fragment (Appendix A, row 2).

For the negative control gene, we used a 443 bp long fragment of pGEM-T easy plasmid (Promega, Cat. A1360, Woods Hollow Road, Madison, WI, USA) for which there is no similar sequence in the *B. terrestris* genome (NCBI; confirmed using Blast searches). We amplified a pGEM-T easy plasmid sequence using costume designed nested primers linked to T7 sequences (Appendix A, row 3). These PCR reactions generated double-strand DNA produce that were separated on agarose gels (0.8%). We excised bands in appropriate sizes, which we cleaned using the Zymoclean gel elution kit (Cat. D4001, Zymo Research, Murphy Ave. Irvine, CA, USA). We generated dsRNA with 1 µg gel-purified PCR products as a template and the RNA Maxx-Transcription Kit (Cat. 200339, Agilent Technologies, Santa Clara, USA) following the manufacturer’s guidelines. We precipitated the dsRNA product and purified it using Lithium Chloride (LiCl) and Ethanol. In brief, 4 M LiCl (0.1 volume) and pre-chilled 100% Ethanol (2.5 volumes) were added to a reaction mixture to precipitate dsRNA for 2 h at −20 °C. We centrifuged the samples at 13,000 g for 15 min at 4^0^ C and washed the pellet with 70% ethanol (100 µL/sample, 1:1 ratio). We resuspended the dsRNA pellet in DNase/RNase free water and assessed the RNA quality (on 0.8% agarose electrophoresis) and quantity (using Nanodrop). The suspended dsRNA was aliquoted and stored at −70 °C until usage. Appendix A summarizes the primers used in all our PCR and RT-PCR reactions.

#### 2.5.1. Perfluorocarbon Nanoparticles

We developed a protocol for loading dsRNA onto perfluorocarbon nanoparticles (PFCnp, [107]) in order to reduce the amounts of injected dsRNA and improve knockdown efficacy. We used a stock solution (20 pmol) comprising 40% (*v*/*v*) perfluorooctyl bromide, 2% (*w*/*v*) safflower oil, 2% (*w*/*v*) of surfactant co-mixture, 1.7% (w/v) glycerin, and rest DNase/RNase free water, courtesy of the lab of Samuel A Wickline (University of South Florida, Health Heart Institute, Fowler Avenue Tampa, FL, USA), which we stored at 4 °C until further use. The freshly prepared PFCnp was measured to gain an average diameter size range of 150–250 nm and a surface charge of +0 ± 20 mV [107,108,109].

#### 2.5.2. Assessing PFC Nanoparticles Toxicity

We serially diluted the PFCnp in ultrapure water while shaking well on rotor shaker and injected (chilled) callow worker bees (8 bees/treatment group) with decreasing volumes of each concentration (Appendix A). The solution was mixed steadily during the whole process of injection in order to keep a consistent distribution of particles. The injected amount was adjusted to the bee body size, as detailed in Table 1. Following injection, we placed the bees for 1–2 min on ice and then transferred them into groups of four of the same treatment into small wooden cages (not controlling for body size distribution across cages). Dead bees were removed each day and the survival rate was determined based on the number of bees alive on Day 7 (Appendix A). We measured the length of the four largest terminal oocytes for all bees alive on Day 7 (Section 2.3). Given that ovarian activity and survival were lower for bees injected with high PFCnp amounts, we chose not to use concentrations above 0.1 pmol in later in-vivo experiments (Appendix A, Kruskal-Wallis; *χ*^2^ = 45.95, df = 7; *p* < 0.0001).

#### 2.5.3. Determining the Efficiency of dsRNA Loading onto the PFCnp

We loaded different concentrations (50–500 µg/mL) dsRNA onto PFCnp at various amounts (0.05–3 pmol), keeping the reaction mixture constant at 50 µL (Appendix A). We slowly (flow rate of ~0.3 mL/minutes) loaded the dsRNA while vortexing the PFCnp (Fine vortex, Korea). We then incubated the dsRNA-PFCnp solution at room temperature for 30 min while rotating it on a shaker (Polymax 1040, Heidolph Instruments) at 200 rpm. We next centrifuged the solution at 10800 g for 10 min at 4^0^ C and collected the supernatant. We used a nanodrop (OD 230/260; ND-1000, Nanodrop Technologies, Inc) reader to estimate the amount of unbound dsRNA in the supernatant. To calculate loading efficiency (LE), we used the following formula: LE% = [1 − (unloaded dsRNA/total dsRNA)] × 100% [110]. These measurements suggest that PFCnp efficiently binds to dsRNA in a dose-dependent manner (Appendix A, Two-way ANOVA, F _(PFC concentration)_ = 578.0, df = 3, *p* < 0.0001; F _(dsRNA amount)_ = 290.2, df = 4, *p* < 0.0001; F _(PFC concentration X dsRNA amount)_ = 24.84, df = 12, *p* < 0.0001).

#### 2.5.4. Determining the Time of PFCnp Injection

We performed two experiments with newly emerged bees for selecting the time and age of injection. In the first experiment, we loaded different dsRNA amounts (50, 100 and 200 µg/mL) onto 0.1 pmol PFCnp and injected the solution to bees at different pair of day combinations (days 2 and 4, 3 and 5, or 4 and 5; days correspond to the age of the bees; Appendix A) and assessed ovarian activity at day 7 of the experiment. For each 2-day combination, we injected bees (*n* = 8) with 1, 2 or 3 µg *Kr-h1* dsRNA. As a control for RNA toxicity, we injected bees with a similar amount of a plasmid dsRNA. We found that both the amount of *Kr-h1* dsRNA loaded onto the nanoparticles and the days of injection affected the ovarian activity (Appendix A). Interestingly, in all three pair-day combinations tested, ovarian activity inhibition diminished with the increase in the amount of loaded dsRNA. We presume that this effect stems from the influence of the RNA on the surface charge of the nanoparticles. Unloaded PFCnps are positively charged, which should facilitate intake into cells; when loaded with dsRNA, the complex becomes more negative, which in high amounts may obstruct intake into cells [111,112,113]. Ovarian activity was lower (i.e., *Kr-h1* knockdown was more effective) for bees injected on days 2 and 4 or 3 and 5 compared to those injected on days 4 and 6 (Appendix A). In contrast, to the clear effect of *Kr-h1* dsRNA, bees injected with pG dsRNA had an ovarian activity similar to control bees, irrespective of the days of injection or dsRNA concentration.

In the second set of experiments, we injected bees with PFCnp loaded with 1 µg dsRNA at different days up to the seventh day post-emergence. The results of this experiment similarly suggest that injections of *Kr-h1* dsRNA on both days 2 and 4, or 3 and 5 produce a better sequence-specific inhibition of ovarian activity compared to the two treatments in which the first injection was on Day 4 (Appendix A).

### 2.6. dsRNA Mediated Kr-h1 Knock-Down

#### 2.6.1. RNA Interference (RNAi) with Naked dsRNA

We first injected 1-day-old callow bees with 5–20 µg naked dsRNA (suspended in water) and used qPCR to measure *Kr-h1* transcript abundance 24–72 h after injection. Given that a single injection failed to produce significant down-regulation with any of the tested doses (data not shown), we switched to a double injection in subsequent experiments (as in Niu et al., 2017, 2016). We randomly assigned freshly collected callow bees to groups of four. On days 2 and 4 after emergence, we injected each bee with 5 µg dsRNA suspended in 5 µL of DNAse/RNAse free ultrapure water (SKU: 01-866-1B, Biological Industries, Beit-Haemek, Israel). We measured *Kr-h1* transcript abundance 3 or 6 h after the second injection on day 4 of the experiment (Appendix A). In a set of complementary experiments, we similarly injected callow bees, and kept them alive for seven days (see experimental outline in Appendix A). We performed daily behavioral observations on days 3 to 5 (see Section 2.2 for details). At the end of the experiment, we measured the amount of wax deposited in each cage and assessed ovarian activity, as described in Section 2.3.

#### 2.6.2. Down-Regulating *Kr-h1* Expression Using dsRNA Loaded on PFCnp

Based on the results of the preliminary experiments (Section 2.5.4), we selected a protocol in which 0.1 µg dsRNA is loaded onto one microliter of PFCnp (0.1 pmol) suspended in water. We injected 10–20 µL PFCnp-dsRNA solution (adjusted to body size as described in Table 1) into the abdomen of focal bees on both day 3 and day 5 post-emergence (Table 1, Figure 1A). As a control for RNA toxicity, we similarly treated bees with PFCnp loaded with similar amounts of pGEMT dsRNA. We collected callow bees (<24 h after emergence from the pupa) from various source (“donor”) colonies (*n* = 5–6 colonies) and cold anesthetized them in 50 mL air-ventilated tubes immersed in ice (~2 °C) until they were immobile for 10–15 min. We measured the length of the front wing marginal cell as an index for body size [114], and classified the bees as “small”, “medium”, or “large” according to the ranges shown in Table 1. We injected different volumes of dsRNA-*Kr-h1* loaded on PFCnp (“*ds-Kr*”), overall delivering 1–2 µg *Kr-h1* dsRNA (according to body size; Table 1). As a control for the effects of dsRNA injection (i.e., “RNA toxicity”), we similarly injected bees with the pGEMT plasmid dsRNA (*“ds-pG”*). An additional control group was used to account for the effect of PFCnp injection. These bees were similarly treated and injected with similar amounts of PFCnp that was not loaded with dsRNA (“*PFC*”). Handling control bees (“*Control*”) were handled and chilled on ice similarly in parallel to bees of the other three groups, but not injected with either dsRNA or PFCnp. Injected bees were left anesthetized on ice (~2 °C) for 2–3 min post-treatment, and were carefully inspected. Bees with visible leakage of the injection solution were discarded and were not used in the experiments. To assess the influence of treatment on gene expression, we flesh-froze bees from all treatment groups in liquid nitrogen, at three, or nine hours after the second injection on Day 5. The bees were dissected, and tissues of interest were stored in an ultra-low freezer (−75–80 °C) until molecular analyses (Section 2.4).

### 2.7. Experiment 1. The Influence of Kr-h1 on JH Regulated Physiology and Behavior in Groups of Similarly Treated Queenless Workers

We repeated this experiment with bees either injected with naked dsRNA or with dsRNA loaded on PFCnp. Naked dsRNA (5 µg/5 µL/bee) was injected on days 2 and 4 after group formation (see experiment outline in Appendix A). Fat bodies and ovaries for qPCR analyses were sampled at 3, and 6 h after the second injection on Day 4. We performed behavioral observations (Section 2.2.) on Day 3 (24 h after the first injection), Day 4 (3 and 6 h after the second injection), and Day 5 (24 h after the second injection). We assessed the amount of wax deposited and ovarian activity on Day seven, as described in Section 2.3.

In a second set of experiments, we injected dsRNA loaded on PFCnp following the procedure described in Section 2.6.2. The outline of this experiment (Figure 1A) is similar to that with the naked RNA, but with the following modifications: First, following the experiments shown in Appendix A, we decided to inject the bees on days 3 and 5, rather than on days 2 and 4 of the experiment. Second, we reduced the dsRNA amount (up to 5-fold, compared to naked dsRNA doses in Appendix A). Third, we collected the brain and fat body tissues for qPCR analyses at 3, and 9 h following the second injection on Day 5.

### 2.8. Experiment 2. The Influence of JH-III on Bees Treated with Kr-h1 dsRNA

The experimental outline (Figure 2A) was overall similar to that in Exp. 1, but following the 1st injection on day 3 of the experiment, half of the bees of all treatment groups were treated with 50 µg/bee JH-III dissolved in Dimethylformamide (DMF), and the other half with only the vehicle (DMF, 3.5 µL/bee). Previous studies showed that treatment with this JH dose stimulates oogenesis, wax deposition, and aggression [69]. The JH (in DMF) or DMF solution alone were topically applied to the thorax. Control bees were similarly handled and chilled on ice, but were not treated with either JH or DMF. All the bees were left on ice 10 min after treatments for maximizing JH absorption. The second dsRNA injection was similar to that described for the first experiment and was done on Day 5.

### 2.9. Experiment 3. The Influence of Kr-h1 on JH Regulated Physiology and Behavior in Groups of Queenless Workers, Each Subjected to a Different Treatment

The experimental outline was similar to that in Exp. 1, but with the following modifications (Figure 3A). First, the bees were treated only once on Day 1, rather than twice, as in Experiment 1. This was done in order to create the groups of bees with normal and reduced *Kr-h1* levels at an early age, assuming the differences at early stages of group formation will affect the establishment of the dominance hierarchy. Secondly, the bees were injected two hours after being collected from the mother colony, rather than Day 3 in the first injection of Exp. 1. Thirdly, following the treatment on Day 1, we assigned the bees to mixed groups. Each one of these groups included one bee subjected to each of the four treatments (Control, PFC, ds-pG, ds-Kr). We aimed to use bees of a similar body size within each cage. Behavioral observations, wax collection, and oocyte length measurements were performed as described for Exp. 1 (Figure 3A and Section 2.2 and Section 2.3).

### 2.10. Experiment 4. The Influence of Kr-h1 on Aggression and Dominance in Groups of Queenless Workers That Have Already Established Dominance Hierarchy

The outline of this experiment is summarized in Figure 4A. We collected newly emerged bees from donor colonies, paint-marked them, and assigned them randomly to groups of four orphan workers. We performed a first set of pre-treatment behavioral observations (Section 2.2) on days 4 and 5 of the experiment and determined the dominance rank for individuals in each queenless group. On Day 5, we injected the α-ranked individual in each group with either *ds-pG* or *ds-Kr* (1–2 µg/bee, corrected for body size) as described in Section 2.6.2. To assess the influence on behavior, we performed behavioral observations during the following days (days 5, 6, and 7 of the experiment; Figure 4A). We assessed ovarian activity and wax deposition on Day 7, as in the other experiments.

### 2.11. Statistical Analyses

We used a one-sample Kolmogorov–Smirnov test to determine whether each set of data are normally distributed. Data that fitted a normal distribution were analyzed using one-way ANOVA followed by Tukey’s post-hoc test. Data that did not fit a normal distribution were analyzed using the non-parametric Kruskal–Wallis-H test followed by Dunn’s post-hoc tests. We used Chi-square tests to compare percentage data (e.g., Figure 3E and Appendix A). Additional information on statistics that were relevant to only a certain experiment are reported in the corresponding methods section or figure legend.

## 3. Results

### 3.1. Experiment 1. The Influence of Kr-h1 on JH Regulated Physiology and Behavior in Groups of Similarly Treated Queenless Workers

We performed this experiment with naked dsRNA, and then after validating the nanoparticles protocol, repeated it with dsRNA loaded on PFCnp. The results using the two methods were similar overall, but the protocol with nanoparticles allowed us to inject lower amounts of dsRNA. Bees injected with naked *Kr-h1* dsRNA, but not with naked control dsRNA (ds-pG), showed a significant reduction of about 50% in fat body and ovarian *Kr-h1* mRNA levels compared to control bees when tested 3 h after the second injection (fat body, Appendix A, One-way ANOVA, F = 6.07, df = 2, *p* = 0.004; ovaries, Appendix A, One-way ANOVA, F = 4.34, df = 2, *p* = 0.018). This influence of *Kr-h1* dsRNA injection was transient and there was no similar reduction observed for bees analyzed six hours after the second injection (Fat body, Appendix A, F = 1.05, df = 2, *p* = 0.356; ovaries, Appendix A, F = 1.25, df = 2, *p* = 0.295). Bees injected with *Kr-h1*, but not with *pG*, dsRNA, deposited less wax in their cages (Appendix A; One-way ANOVA F = 9.982, df = 2, *p* = 0.0004), and had less active ovaries (Appendix A; Kruskal–Wallis test, *χ*^2^ = 35.75, df = 2, *p* < 0.0001) compared to control bees. Injection of naked *Kr-h1* dsRNA also affected behavior. Compared to handling control bees and those injected with pG dsRNA, the bees injected with naked *Kr-h1* dsRNA showed fewer threatening displays when observed at three, but not at six hours, after the second injection (Appendix A; Kruskal–Wallis test, *χ*^2^*_(threatening display, 3 after)_* = 9.93, df = 2, *p* = 0.007; *χ*^2^*_(threatening display, 6 after)_* = 0.34, df = 2, *p* = 0.843, respectively). The number of threatening displays were not affected by treatment in the two other observation sessions (Appendix A, *χ*^2^*_(threatening display, 24 before)_* = 0.71, df = 2, *p* = 0.701; *χ*^2^*_(threatening display, 24 after)_* = 0.75, df = 2, *p* = 0.756).

The experiment in which we treated bees with dsRNA loaded onto PFCnp produced similar transient *Kr-h1* down-regulation in the fat body that was seen at 3, but not at 9 h after dsRNA injection on Day 5 (Figure 1B; Kruskal–Wallis, *χ*^2^*_(Fat bodies,3 h)_* = 27.90, df = 3, *p* < 0.0001; *χ*^2^*_(fat bodies,9 h)_* = 7.70, df = 3, *p* = 0.052, respectively). In this experiment, we also measured *Kr-h1* levels in the brain and found a similar transient downregulation (Figure 1C; Kruskal–Wallis, *χ*^2^*_(Brain,3 h)_* = 25.71, df = 3, *p* < 0.0001; *χ*^2^*_(Brain,9 h)_* = 5.28, df = 3, *p* = 0.15, respectively). In a set of complementary experiments, we tested the influence of *Kr-h1* knockdown on JH regulated behavior and physiology. We found that bees injected with PFCnp loaded with *Kr-h1* dsRNA showed significantly reduced ovarian activity compared to bees from all three control groups (Figure 1D, Kruskal–Wallis *χ*^2^ = 13.17, *df* = 3, *p* = 0.004). An overall similar pattern was seen for wax deposition, but the differences compared to the PFCnp and *ds-pG* treatment groups were not statistically significant in the post-hoc analyses (Figure 1E, One-way ANOVA, *F* = 3.94, *df* = 3, *p* = 0.018). *Kr-h1* down-regulation also influenced the number of threatening displays recorded during the observations performed 3, 6 and 24 h after the second injection. Bees injected with *Kr-h1* dsRNA performed fewer threatening displays (Figure 1F; Kruskal–Wallis test*; χ*^2^ = 17.69, *df* = 3, *p* = 0.0005; *χ*^2^ = 12.97, *df* = 3, *p* = 0.005; *χ*^2^ = 9.9, *df* = 3, *p* = 0.019; for bees observed 3, 6, or 24 h. post the 2nd injection, respectively). Post-hoc analyses for the observations performed at 3 and 6 h after the second injection revealed a statistically significant reduction in threatening display in bees treated with *Kr-h1* dsRNA compared to bees in the control and PFC treatment groups, and a similar but statistically non-significant trend compared to the *ds-pG* treated bees. The number of threatening displays was significantly reduced in the *Kr-h1* treated bees compared to the PFC, but not the Control or ds-pG treatments, at the observations performed 24 h after the second injection. Nevertheless, the overall trend was similar at the three post 2nd injection observations. Taken together, the experiments with both naked and PFCnp loaded dsRNA suggest that *Kr-h1* mediates the influence of JH on ovarian activity, wax deposition, and agonistic behaviors.

### 3.2. Experiment 2. The Influence of JH on Bees Treated with Kr-h1 dsRNA

In order to test whether JH can act independently of *Kr-h1*, we measured wax deposition and ovarian activity for bees treated with JH right after the first *Kr-h1* dsRNA injection. Controlled bees from each one of the RNAi treatments groups (i.e., Control, PFC, ds-pG, ds-Kr) were similarly treated, but were topically treated with only the vehicle (DMF). Given that there were two trials, we first performed a 3-way ANOVA using the square root transformed data (that fit normal distribution; Kolmogorov–Smirnov test, *p* > 0.05) with RNAi treatment, JH treatment, and trial as factors (Appendix A). Given that these analyses showed no significant effect for the trial, on either ovarian activity (*p* = 0.36), or wax deposition (*p* = 0.11), we reanalyzed the data using 2-way ANOVA followed by Holm–Sidak post-hoc tests (Figure 2). These analyses revealed a significant effect of RNAi treatment for both ovarian activity (Figure 2B; Two-way ANOVA, F = 6.987, df = 3, *p* < 0.001) and wax deposition (Figure 2C; F = 16.536, df = 3, *p* < 0.001). JH treatment had an overall significant effect on ovarian activity (Figure 2B; F = 11.238, df = 1, *p* = 0.001), but not on wax deposition (Figure 2C; F = 2.178, df = 1, *p = 0.148*). The interaction between the RNAi and JH treatments was not statistically significant for either ovarian activity (Figure 2B; Two-way ANOVA, F = 0.08, df = 3, *p = 0.97*), or wax deposition (Figure 2C; Two-way ANOVA, F = 0.235, df = 3, *p* = 0.87). Bees treated with JH-III had more active ovaries compared with bees treated with only the DMF vehicle. However, JH treatment could not fully recover the inhibitory effect of *Kr-h1* downregulation on either ovarian activity or wax production (Figure 2B,C). Taken together, this experiment shows that a JH treatment that is sufficient to induce ovarian activity cannot fully recover the effect of down-regulating *Kr-h1* expression levels. These findings are consistent with the premise that *Kr-h1* mediates the influence of JH on ovarian activity and wax deposition. The trend of decrease in ovarian activity and wax deposition in bees treated with both *Kr-h1* dsRNA and JH can be explained by the partial and transient effect of *Kr-h1* down-regulation (Figure 1 and Appendix A).

### 3.3. Experiment 3. The Influence of Kr-h1 on JH Regulated Physiology and Behavior in Groups of Queenless Workers, Each Subjected to a Different Treatment

Given that dominance rank is the product of interactions between individuals in a group, studies in which all the individuals are subjected to the same treatment (such as Exp. 1) may not reveal the full effect of *Kr-h1* on dominance. To fill this gap, we performed an additional experiment in which each bee in a queenless group was subjected to a different treatment. The outline of this experiment is summarized in Figure 3A. Consistent with Exp. 1, treatment had a significant influence on ovarian activity (Figure 3B; Kruskal–Wallis *χ*^2^ = 19.20, *df* = 3, *p* = 0.0002). Ovarian activity was the lowest for bees treated with *Kr-h1* dsRNA, but the post-hoc comparison was statistically significant only relative to the Control and *ds-pG* treated bees. In a pooled analysis of the three observations performed during days 3–5, we found that bees injected with *Kr-h1* dsRNA showed fewer threatening displays compared to bees of the Control and *ds-pG* treatment groups, but not compared to the PFCnp treatment (Figure 3C; Kruskal–Wallis test, *χ*^2^ = 21.00, *df* = 3, *p* = 0.0001). Bees treated with *Kr-h1* dsRNA also had low dominance index (Figure 3D, Kruskal–Wallis *χ*^2^ = 23.48, *df* = 3, *p* < 0.0001) and had significantly lower dominance rank (Figure 3E; contingency table analysis using 4X4 chi-square test, *χ*^2^ = 152.64, *p* <0.00001, followed by pair-wise Chi-square tests, and corrected for multiple comparisons). The differences in dominance index were statistically significant compared to the Control and *ds-pG* treated bees, but not compared to the nanoparticles control treatment (PFCnp; Figure 3D). The results of the behavioral analyses (threatening displays, dominance index, and dominance rank) were similar when we analyzed each observation session separately (days 3, 4, and 5; Appendix A).

### 3.4. Experiment 4. The Influence of Kr-h1 on Aggression and Dominance in Groups of Queenless Workers That Have Already Established a Dominance Hierarchy

Given that Experiments 1 and 3 established that *Kr-h1* influences the establishment of dominance hierarchies, we next asked whether it is also needed for maintaining a high dominance rank in established dominance hierarchies. To meet this goal, we knockdown *Kr-h1* expression in the most dominant individuals (α rank) in groups that have already established stable dominance hierarchies (see experimental outline in Figure 4A). We found that α-rank bees injected with dsKr-h1 on Day 5 showed an ovarian activity similar to that of control α-rank bees injected with ds-pG when analyzed two days later (Figure 4B, Two-tailed, *t*-test, *df* = 8, *p* = 0.412). The frequency of threatening displays performed by the α-rank bee decreased significantly after *ds-Kr* dsRNA but not *ds-pG* dsRNA injection (Figure 4C shows the average for 30 min of the two observations before injection compared to the average of the three observations performed after dsRNA injection: Two-tailed, *t*-test, *df* = 8, *p* = 0.018, *p* = 0.445, respectively). In a complementary Two-way ANOVA with observation time (before or after dsRNA injection) and dsRNA treatment (*Kr-h1* or pG), we found that the level of threatening displays is significantly lower for the bees injected with *Kr-h1* dsRNA compared to the three other treatment groups (Figure 4C; Two-way ANOVA, F = 4.90, *_(Treatment)_*_,_ df = 1, *p* = 0.041; F = 7.72*_(Observation)_*, df = 1, *p* = 0.013; F = 2.99 *_(Treatment x observations),_* df = 1, *p* = 0.103, Bonferroni post-hoc test). Consistent with the effects on the frequency of threatening displays, *Kr-h1* knockdown also affected dominance rank: α rank bees injected with *Kr-h1* dsRNA were more likely to lose their top dominance rank compared to α rank bees injected with pG-dsRNA (Figure 4D, Chi-square with Yates’ correction = 31.69, df = 1, *p* < 0.0001). Nevertheless, it is notable that in all the groups they did not have a rank lower than Betta.

## 4. Discussion

High social dominance rank has fitness advantages and a genetic bases, but little is known about the specific genes and molecular processes influencing dominance rank [9,115]. We know particularly little about the genes influencing dominance in females and in invertebrates. Bridging these gaps in knowledge is crucial for articulating the molecular organization principles governing dominance behavior and defining their generality across sex and phylogeny. To address his challenge, we developed and validated a new nanoparticle-based dsRNA-mediated RNAi protocol and used this procedure to down-regulate the expression of *Kr-h1* in newly emerged adult bumble bee workers. The development of this novel protocol is an important contribution because it allows knocking down gene expression in a bumble bee using relatively low amounts of dsRNA, and reduced RNA toxicity and nonspecific effects. Using this approach, we showed that *Kr-h1* mediates not only the effects of JH on fertility and wax deposition, but also on agonistic behavior and dominance rank. By showing that bees with reduced *Kr-h1* transcript abundance show less dominance behavior and have a lower dominance rank compared to bees similarly injected with control dsRNA, we establish for the first time a causal link between a specific gene and dominance rank in a social insect in which dominance hierarchies are established between females. These findings provide a molecular link between JH and dominance and sets the stage for research on the molecular and neuronal bases of dominance in social insects and other female hierarchies.

RNAi technology in insects has faced challenges related to the stability, delivery, and knock-down effectivity of dsRNA/siRNA [116,117]. dsRNA that is not protected in the insect hemolymph is typically degraded and therefore injected dsRNA can be cleared from the hemolymph before effectively knocking down the targeted gene [118,119]. Indeed, our unpublished in-vitro degradation assay for dsRNA in bumble bee hemolymph showed that most of the tested dsRNA is degraded within about six hours. Nanoparticles offer a system to improve delivery, uptake, and efficiency (reduction in the amount of dsRNA injected). Nanoparticles may also help in protecting the introduced dsRNA from rapid degradation [120,121]. It is important to note that the *B. terrestris* genome lacks a *Sid 2* gene, which is an important component of systemic RNAi signaling. There is evidence that the *Sid-1/2* genes facilitate dsRNA uptake into cells and therefore improve the effectivity of RNAi [122,123]. Thus, the lack of *Sid2* means that the development of RNAi technologies in this species is expected to be even more challenging [124]. Our validated PFCnp RNAi protocol enables us to significantly reduce (~5 fold) the amount of dsRNA injected, which is important for decreasing dsRNA toxicity and nonspecific effects. However, it should be noted that some of our experiments suggest that unloaded PFCnp may have some biological effects, perhaps due to the positive charge of their surface.

In vertebrates in which the physiological and molecular mechanisms affecting dominance behavior are better understood, particularly in males, it is established that the sex steroid hormones that coordinate processes related to reproduction also influence dominance [125,126]. JH seems to overall have similar coordinating functions in females of simple (“primitive”) insect societies [65,69,127,128,129,130,131]. The bumble bee *B. terrestris* provides some of the best support for this notion. Endocrine manipulation experiments show that JH coordinates processes associated with reproduction in many tissues. These include ovary activation, wax deposition, regulation of biosynthetic pathways in exocrine glands, and gene expression in the brain and fat body [47,70]. Similar to vertebrates’ sex steroids, JH also influences behaviors such as aggression, dominance, and circadian rhythms [69,74]. The new findings presented above show that worker bees with down-regulated *Kr-h1* mRNA levels are overall similar to bees with naturally or artificially reduced JH titers. Bees with knocked down *Kr-h1* levels have smaller ovaries (Figure 1D, Figure 2B, Figure 3A, Appendix A and Appendix A), deposited less wax in their cages (Figure 1E, Figure 2C and Appendix A), and showed fewer threatening displays, compared to bees similarly injected with control dsRNA. These findings are consistent with evidence that *Kr-h1* is a JH responsive gene in *B. terrestris* [47,83]. Our gene knock-down experiments establish *Kr-h1* as a pivotal signaling gene that is expressed in multiple tissues and acts downstream of JH in bumble bees, consistent with its key roles in JH signaling pathways in other insect species [82,86,89,94,95,96,97,132,133,134,135,136]. These findings also set the stage for a deeper study on JH signaling pathways and the genes with which *Kr-h1* interacts in bumble bees, as well as other hymenopteran insects.

The effects of *Kr-h1* knock-down on diverse processes such as oogenesis, wax deposition, and behavior are consistent with the premise that *Kr-h1* mediates the influence of JH by modulating gene expression in diverse tissues including the fat body, wax glands and brain. Our qPCR measurements are consistent with this premise by showing that bees injected with *Kr-h1* dsRNA show reduced *Kr-h1* RNA abundance in the brain, fat body, and ovaries (Figure 1B,C, and Appendix A). The effects of *Kr-h1* dsRNA injection were substantial, albeit our qPCR mRNA measurements suggest that *Kr-h1* down-regulation was partial and transient: *Kr-h1* transcript abundance was significantly reduced when measured three hours post injection, but not a few hours later (Figure 1B,C and Appendix A). This transient RNA knockdown is consistent with evidence suggesting that modulation of JH amounts in newly emerged adult bees is sufficient to produce lasting effects, including on ovarian activity measured about a week post JH treatment [68,69,70,74,83,137]. Similar long-lasting effects of single early JH manipulations have been also reported for other insects [138,139,140,141,142,143]. Given that JH is not stable in the hemolymph over time [74,144,145,146,147,148], the lasting effects of a single acute JH treatment early in life is consistent with organizational effects, or with models in which JH levels at early age regulate cascades of processes that after the initial activation do not require continuous high JH levels. Our current finding suggests that many of these effects are mediated by *Kr-h1*.

As in other species, the establishment of dominance hierarchies among bumble bee workers is achieved through repeated agonistic interactions characterized by frequent threatening displays [69,72,101,102,104]. Our findings lend credence to the hypothesis that *Kr-h1* mediates the influence of JH on dominance in bumble bees, providing the first causal link between a specific gene and dominance in social insects. *Kr-h1* mRNA levels are higher in the brain of dominant compared to low-ranked individuals [83] and our results show that our RNAi protocol significantly reduced *Kr-h1* in the brain (Figure 1). These results are consistent with the premise that *Kr-h1* mediates JH effects on dominance by modulating gene expression in the brain tissue. However, it is also possible that *Kr-h1*-mediated JH effects in other tissues (such as the ovaries) lead to physiological changes that influence dominance. For example, JH may affect metabolism that can affect fighting ability and RHP. Moreover, JH activation of the ovaries may lead to increase production of ecdysteroids [103] or other endocrine signals that may act on the brain. Worker bees with down-regulated *Kr-h1* mRNA levels showed fewer threatening displays towards other individuals in all the experiments in which we performed behavioral observations. Notably, in most hierarchies in the mix treatment experiment, the dsRNA injected bees had the lowest (*δ*) or next to lowest (*γ*) dominance rank, which was significantly lower than for bees injected with control dsRNA (Figure 3D). Top-ranked α bees in already established hierarchies of older bees for which we knocked down *Kr-h1* levels showed reduced threatening displays compared to the period before dsRNA injection and compared to similar α bees injected with control dsRNA and were more likely to lose their top dominance rank (Figure 4). These later results may suggest that *Kr-h1* is involved not only in the establishment but also in the maintenance of dominance hierarchies, but additional research is needed for testing this hypothesis.

In spite of the overall agreement, some of our results suggest that *Kr-h1* knockdown did not always mirror the effect of reducing JH levels. Dominant bees in already established dominance hierarchies that were treated with Precocene-I to reduce circulating JH levels, showed significantly lower ovarian activity, but not a lesser amount of threatening displays compared to similar bees treated with only the vehicle [69]. In contrast, here we show that similar bees treated with *Kr-h1* dsRNA performed fewer threatening displays and were more likely to lose their top dominance rank but had similarly active ovaries compared to bees treated with control dsRNA. Although we cannot exclude the possibility that these differences stem from technical or biological (e.g., genetic, or developmental) variability between the two experiments, it is worth noting that recent studies with other insects show that JH reduction and *Kr-h1* knockdown do not necessarily have identical effects. JH acts on additional transcription factors other than *Kr-h1* and has various epigenetic effects that may regulate the activity of *Kr-h1* or other downstream processes [91,133,149,150,151,152,153]. In the future, it will be important to explore whether additional factors other than JH affect *Kr-h1* activity and its effects on dominance. Likewise, it is important to test whether JH influences dominance by acting on molecular processes other than those regulated by *Kr-h1*.

## 5. Conclusions

Although it is well established that dominance rank has significant fitness consequences across the animal kingdom, it has been proven difficult to study in molecular terms. Given its complexity and strong context dependency (including the traits of individuals with whom the focal subject interacts with), a comprehensive understanding of dominance behavior requires studying diverse social systems. Our study provides the first causal link between a specific gene and dominance rank in female hierarchies of insects. Although additional studies are needed for determining the generality of our findings, they may suggest a common theme across species and gender by which key genes in endocrine signaling pathways controlling reproduction also affect dominance. The implication of *Kr-h1* in the regulation of dominance in bees sets the stage for studies on the molecular, neurobiological, and anatomical underpinnings of social dominance in insects and other invertebrates as well as in female social hierarchies. This provides an important step towards articulating general models for the genetic and molecular regulation of dominance hierarchies. Our new nanoparticle-based RNAi protocol allows us to knockdown gene expression with relatively low dsRNA amounts and reduced RNA toxicity and nonspecific effects. Abdominal injection of our dsRNA nanoparticle complex enables *Kr-h1* mRNA knockdown not only in abdominal tissues such as the fat body and ovaries, but also in the brain. 

## Figures and Tables

**Figure 1 biology-10-01188-f001:**
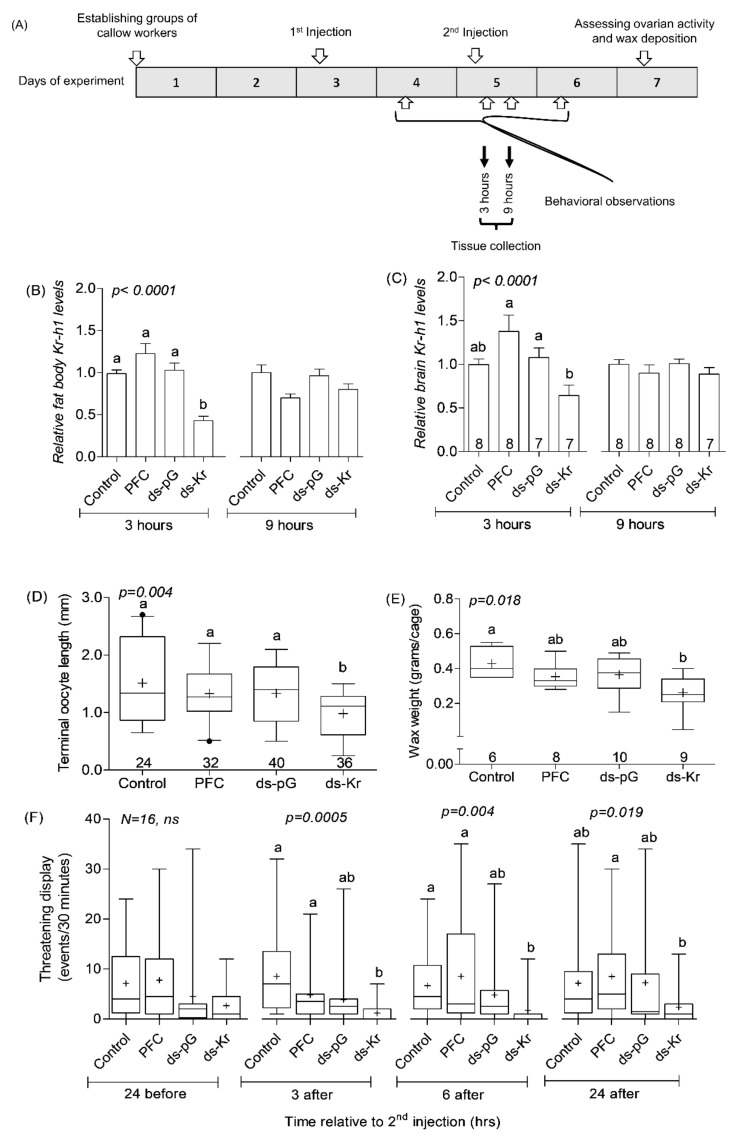
**The influence of nanoparticles bounded *Kr-h1* dsRNA on gene expression, physiology, and behavior.** (**A**) Experiment outline. (**B**) Fat body *Kr-h1*, mRNA levels. (**C**) Brain *Kr-h1*, mRNA levels. *Kr-h1* mRNA levels were measured for bees sampled at 3 or 9 h after the second injection. (**D**) Ovarian activity. (**E**) Wax deposited in the cage. (**F**) Threatening displays. Control—untreated bees, PFC—bees injected with nanoparticles not loaded with dsRNA, ds-pG—bees injected with *pG* dsRNA, ds-Kr—bees injected with *Kr-h1* dsRNA. In (**B**,**C**,**F**), a separate statistical analysis was performed for each time point. The sample size reported within bras in panel (**B**) are also applicable for panel (**C**). The box plot presents the median (−) and mean (+) values, with the frame spanning over the first to the third quartile; the whiskers depict the 5th/95th percentile; outliers are depicted with black dots. Treatments marked with different small letters were statistically different in a Kruskal-Wallis test followed by Dunne’s post-hoc analysis (**D**,**F**) or ANOVA followed by Tukey’s post-hoc test (**E**). Sample size depicts the number of bees in panels (**B**–**D**,**F**), and the number of cages (i.e., groups) in panel (**E**).

**Figure 2 biology-10-01188-f002:**
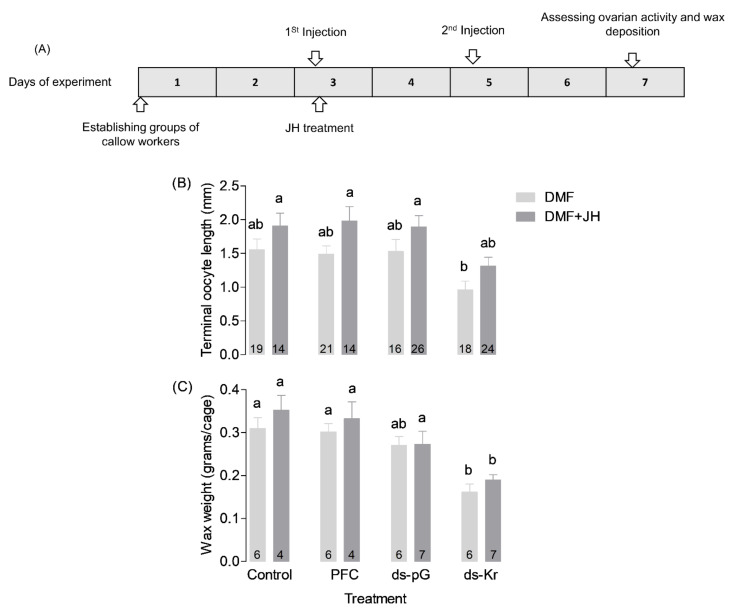
Treatment with JH-III could not fully recover the effect of Kr-h1 down-regulation. (**A**) Experimental outline. (**B**) Ovarian activity at seven days of age. (**C**) Total amount of wax deposited. Statistical analysis for the length of terminal oocyte was performed using square root transformed data, which fits normal distribution. Treatments marked with different small letters are statistically different in Two-way ANOVA followed by Holm–Sidak post-hoc tests. Numbers within bars show sample size, which is the number of bees in (**B**), and the number of cages in (**C**). Further details are shown in Figure 1.

**Figure 3 biology-10-01188-f003:**
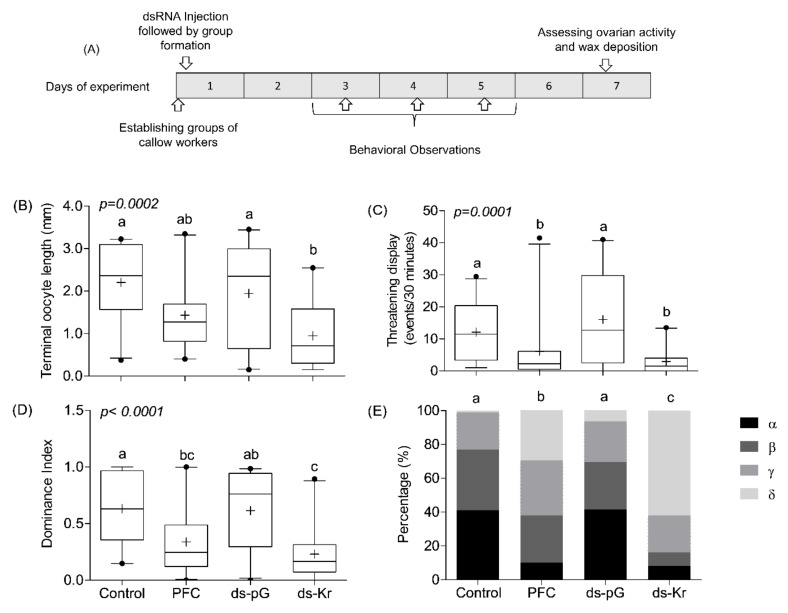
The influence of *Kr-h1* down-regulation on dominance and aggression in mixed treatment groups. (**A**) Experimental outline. (**B**) Ovarian activity at seven days of age. (**C**) Threatening displays. (**D**) Dominance index. (**E**) Proportion of dominance rank. The data shown in panel (**C**–**E**) are the average of the observations of days 3, 4, and 5; an overall similar pattern was obtained when each day was analyzed separately (Appendix A). Treatments marked with different small letters in panels (**B**–**D**) are statistically different in a Kruskal–Wallis H test followed by Dunn’s post-hoc test, and in Chi square tests in panel (**E**) (see text for details). Sample size is *n* = 22 for all analyses. Further details are shown in Figure 2.

**Figure 4 biology-10-01188-f004:**
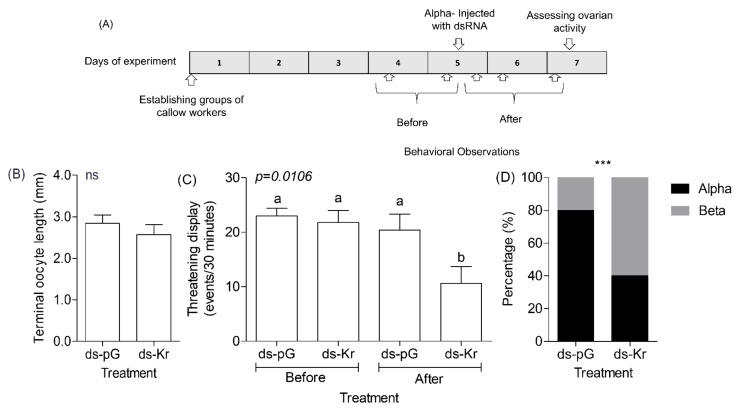
The influence *Kr-h1* knock-down on behavior and ovarian activity of dominant queenless bees in established dominance hierarchies. (**A**) Experimental outline. (**B**) Ovarian activity at seven days of age of dominant (alpha) bees injected with *Kr-h1* dsRNA (*ds-kr*) or control dsRNA (*ds-pG*). (**C**) Threatening displays of dominant bees injected with *Kr-h1* or control dsRNA before or after injection; the values before injection are the average of the observations in days 4 and 5, and the values after injection are the average of the observations in days 5 (after injection), 6, and 7. Treatments marked with different small letters are statistically different in two-way ANOVA analysis followed by Bonferroni post-hoc analysis. (**D**) Dominance rank of alpha bees after injection with either *Kr-h1* or pG dsRNA. The vertical bars (in **B**,**C**) depict mean ± SE. *n* = 5 bees per treatment group. ***—*p* < 0.0001, in Chi-square with Yates’ correction.

**Table 1 biology-10-01188-t001:** Injection volumes and RNA amounts injected into bees of different body sizes.

Body Size Category	Marginal Wing Cell Length (mm)	Injection Volume (µL/Bee)	dsRNA Amount (µg/Bee)
Small	2.6–2.8	10	1.0
Medium	2.9–3.0	15	1.5
Large	3.1–3.3	20	2.0

## Data Availability

Not applicable.

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
