# Peer review of "Krüppel-homologue 1 Mediates Hormonally Regulated Dominance Rank in a Social Bee"

_biology, 2021, doi:10.3390/biology10111188_

Round 1

Reviewer 1 Report

Manuscript Number: Biology 1401796 . Krüppel-homologue 1 mediates hormonally-regulated dominance rank in a social insect.

The authors studied the mechanism of social dominance by testing the hypothesis that the gene Kr-h1, a transcript factor, mediates via JH dominance in queenless workers of Bombus terrestris. The test is performed with the knock-down approach with RNAi. For knocking down Kr-h1 expression the authors have developed an improved protocol, which is an impressive work, as could be read in the Methods (rule 175-359). This means that the manuscript have two topics. In the Results these are combined, as is the research is done with the new protocol, however, making it more complex to distract the answer on the research question.

In the Introduction is a misbalance in the background information on the two topics. A very long (too long) introduction about dominance and Kr-h1 (rule 28-150), and only a few sentences about the improved protocol (rule 154-156). Some information about several aspects in the research are missing as clarified below. Some information is not relevant for the scope of the manuscript.  

The manuscript needs a major revision concerning the structure, and on relevant and irrelevant information.

Introduction

r29. The authors defined dominance as the outcome of contests between pairs of conspecifics. To complete the definition it needs to be included that the contests must be multicontextual and the outcome unidirectional.

r37. It is not clear why the authors come up here with “genetically inherited”, while the proximate cause of dominance is not explained yet. One of the proximate factors resulting is the extrinsic self-reinforcing winner-loser effect (more or less indicated in r44). In mammals explained by the hormones testosterone and cortisone. A second factor is the subjective evaluation of a resource, including motivation. Another factor is family (matrilines) and/or network of allies. The last factor, which explains mostly the dominance, is the intrinsic Resourse Holding Potential (RHP) or prior attributes. This attributes are size, weight, physical strength/ fighting ability, age, sex, in which genes and hormones are involved.  RHP is the topic of this study, although not named in such way. In r48-51 the authors mention the involvement of genes. A more structural explanation is needed.

r52-71 and r99-102 The authors explain broadly the genes and hormones in mammals, what is less relevant for the topic of this manuscript, and could be leaved out.

r89-90. Why the emphasis on “unmated workers” instead of queenless workers? In the context of this paper the “laying of unfertilized haploid eggs that develop in males” is not relevant.

r93-98. “…..certain neurogenetics patterns…..” “……..differential expression of some the same genes….”. This needs explanation or it must be leaved out.

r131. Out of the blue the authors come up with that the hypothesis that Krupple-homolog 1 influences dominance rank. R 132-141 is not relevant.

r142-150 This part can be more elaborated to make more clear why Kr-h1 should be involved in dominance. Can the authors describe a pathway, including CA , ovarian activity and in which tissues Kr-h1 is active, as why the fat body is used? Ending with the hypothesis that Krupple-homolog 1 influences dominance rank.

r154-156 The authors use words/terms as dsRNA, Perfluorocarbon nanoparticles, without further explanation. More explanation is needed, or the authors can leave this out of the Introduction and replace it to the Material and Methods. Material and Methods could be started with 1. Improvement of the protocol with 1.2-1.4,  with headings about the function, like knock down efficacy, and control on toxicity and so on. 2. Experiments with Kr-h1 and JH with 2.1 Bees, 2.2 Behavioural observation and so on.

In the Introduction the function of wax production (wax deposition is a parameter in this research) in relation to dominance is not explained. For fertility, the authors use ovarian activity. Why the authors have not used also egg laying, which would the ultimate reason for a high dominance rank in bumble bees? In bumble bee workers the development of ripe eggs and laying of these eggs are separated processes.

Methods

 r162-173  There is no information about the forming of callow groups: the age of the callow workers in hours, number of queenless workers in a group, how many groups, the statistics per group or all workers, how is corrected for difference between the workers per group due to rank differences.

The authors must explain why is chosen for day 7 to determine ovarian activity.

r351 The retreats and the total number of encounters is used for the calculation of DI. It needs to be clarified what the encounters are (resulting behaviours as buzzing, pumping, darting, attacks?). The retreat behaviour is not described! Why is retreat behaviour used for the dominance rank? This needs to be clarified.

Results

A lot of results, with many graphs in the paper and supplement.

r533. Make it the reader easier by indicating what the numbers are meaning: in figure 1B the number of bees, regardless the groups; in figure 1C the number of groups? The same for the other figures.

It seems that there is much more variability in the control and ds-pG control, especially in figure 3. Can the authors explain that?

Discussion:

r617-619, 625 can be leaved out (see comments introduction). It is not relevant to compare is with the sex steroids in vertebrates.

r668 The authors indicate “the first causal link between a specific gene and dominance in social insects”.  Is the gene directly involved in behaviours resulting in dominance (more discussion about dominance in queenless worker is needed including RHP)? What about the relation with ovarian activity and wax deposition. More discussion is needed as Kr-h1 is described as an early juvenile hormone-response gene. Are there suggestions for a pathway?  Is vitellogenin involved? The Kr-h1 down regulation in brain and fat body is not discussed (?)

The improved protocol is not discussed, as a large part of the paper concerns this topic.

Author Response

The authors studied the mechanism of social dominance by testing the hypothesis that the gene Kr-h1, a transcript factor, mediates via JH dominance in queenless workers of Bombus terrestris. The test is performed with the knock-down approach with RNAi. For knocking down Kr-h1 expression the authors have developed an improved protocol, which is an impressive work, as could be read in the Methods (rule 175-359). This means that the manuscript have two topics. In the Results these are combined, as is the research is done with the new protocol, however, making it more complex to distract the answer on the research question.

In the Introduction is a misbalance in the background information on the two topics. A very long (too long) introduction about dominance and Kr-h1 (rule 28-150), and only a few sentences about the improved protocol (rule 154-156). Some information about several aspects in the research are missing as clarified below. Some information is not relevant for the scope of the manuscript. 

The manuscript needs a major revision concerning the structure, and on relevant and irrelevant information.

*** - The reviewer is correct that there are two main contributions in our paper. We indeed struggled quite a lot on how to balance the weight given to the technical innovation (the development of the PFCnp protocol, in which we invested a great amount of time, efforts and resources) and the conceptual breakthrough (identifying a first gene for dominance in a social insect). Eventually, we decided to focused our contribution on a gene for dominance, and submit it to a multidisciplinary journal such as Biology. Now, given the reviewer comments we significantly extended the space and attention to the technical innovation of our paper and refer to it in the Discussion as well as in other parts of the paper. This include adding a new Discussion paragraph summarizing the technical contribution of our work (lines # 670-686).

Specific comments:

Introduction

r29. The authors defined dominance as the outcome of contests between pairs of conspecifics. To complete the definition it needs to be included that the contests must be multicontextual and the outcome unidirectional.

*** - We elaborated the dominance definition along with the reviewer's suggestion (now lines # 39-42).

r37. It is not clear why the authors come up here with “genetically inherited”, while the proximate cause of dominance is not explained yet. One of the proximate factors resulting is the extrinsic self-reinforcing winner-loser effect (more or less indicated in r44). In mammals explained by the hormones testosterone and cortisone. A second factor is the subjective evaluation of a resource, including motivation. Another factor is family (matrilines) and/or network of allies. The last factor, which explains mostly the dominance, is the intrinsic Resourse Holding Potential (RHP) or prior attributes. This attributes are size, weight, physical strength/ fighting ability, age, sex, in which genes and hormones are involved.  RHP is the topic of this study, although not named in such way. In r48-51 the authors mention the involvement of genes. A more structural explanation is needed.

*** - We thank the reviewer for this clarification which helped us better articulate this part of the Introduction, which we rewrote to address his comment. However, we do not agree with the reviewer view that hormones such as JH and genes are limited to RHP. Hormones and genes can also influence learning capacity, sensory processes, and motivation, which are important for dominance hierarchy interactions. Some evidence for this is mentioned later in our Introduction (lines # 55-61).

r52-71 and r99-102 The authors explain broadly the genes and hormones in mammals, what is less relevant for the topic of this manuscript, and could be leaved out.

***- Given that we submit the paper to a journal with a broad readership, and not limited to entomologists, we think that it is important to keep a broader context and refer to what is known in vertebrates, which have been studied more extensively, and therefore can provide excellent framework for understanding the importance of our study. Nevertheless, given the reviewer’s comment, we shorten the reference to specific studies with vertebrates. 

r89-90. Why the emphasis on “unmated workers” instead of queenless workers? In the context of this paper the “laying of unfertilized haploid eggs that develop in males” is not relevant.

***- We believe that this background on research with social insects is crucial for understanding the context of our research, as well as our specific study system (now lines # 100-101). 

r93-98. “…..certain neurogenetics patterns…..” “……..differential expression of some the same genes….”. This needs explanation or it must be leaved out.

***- Fixed (line # 105).

r131. Out of the blue the authors come up with that the hypothesis that Krupple-homolog 1 influences dominance rank. R 132-141 is not relevant.

***- We rewrote and elaborated this part and hope that our decision to focus on Kr-h1 is now well explained (lines # 142-147).

r142-150 This part can be more elaborated to make more clear why Kr-h1 should be involved in dominance. Can the authors describe a pathway, including CA , ovarian activity and in which tissues Kr-h1 is active, as why the fat body is used? Ending with the hypothesis that Krupple-homolog 1 influences dominance rank.

***- Modified (Lines # 142-154, 163-171, and 732-336)

r154-156 The authors use words/terms as dsRNA, Perfluorocarbon nanoparticles, without further explanation. More explanation is needed, or the authors can leave this out of the Introduction and replace it to the Material and Methods. Material and Methods could be started with 1. Improvement of the protocol with 1.2-1.4,  with headings about the function, like knock down efficacy, and control on toxicity and so on. 2. Experiments with Kr-h1 and JH with 2.1 Bees, 2.2 Behavioural observation and so on.

***- We explained the unclear terms and removed some methodological information along with the reviewer’s suggestion (lines # 172-180).

In the Introduction the function of wax production (wax deposition is a parameter in this research) in relation to dominance is not explained.

*** we now refer the wax deposition and other fertility related processes when first mentioning JH in the bumble bee (lines # 125-128).

For fertility, the authors use ovarian activity. Why the authors have not used also egg laying, which would the ultimate reason for a high dominance rank in bumble bees? In bumble bee workers the development of ripe eggs and laying of these eggs are separated processes.

***- We do not agree with the reviewer that egg-laying and oogenesis are separated processes. The Reviewer is right that ripe eggs are not necessarily laid, but worker with inhibited ovarian activity do not have ripe eggs and therefore cannot lay eggs forcing a strong link between the two processes. An additional reason that we chose to focus on ovarian activity rather than on egg laying is that oogenesis is related to vitellogenesis and regulated by JH. On the other hand, very little is known on the endocrine regulation of egg-laying (there is some evidence consistent with a role for ecdysteroids, but additional studies are needed to support this hypothesis; see our COIS 2015 review).

Methods

 r162-173  There is no information about the forming of callow groups: the age of the callow workers in hours, number of queenless workers in a group, how many groups, the statistics per group or all workers, how is corrected for difference between the workers per group due to rank differences.

***- We added the requested information (lines# 194-197).

The authors must explain why is chosen for day 7 to determine ovarian activity.

***- We added this information (lines 229-232).

r351 The retreats and the total number of encounters is used for the calculation of DI. It needs to be clarified what the encounters are (resulting behaviours as buzzing, pumping, darting, attacks?). The retreat behaviour is not described! Why is retreat behaviour used for the dominance rank? This needs to be clarified.

***- We added the requested information (lines 209-214).

Results

A lot of results, with many graphs in the paper and supplement.

r533. Make it the reader easier by indicating what the numbers are meaning: in figure 1B the number of bees, regardless the groups; in figure 1C the number of groups? The same for the other figures.

***- Done (lines # 569-570).

It seems that there is much more variability in the control and ds-pG control, especially in figure 3. Can the authors explain that?

***- Variation seems to be smaller in the ds-Kr groups probably because all bees subjected to this treatment have down regulated JH signaling and therefore undeveloped ovaries and lower aggression. The various control groups are expected to be more variable – some individuals may have low and others high JH signaling. The PFC treatment in Exp. 3 seemed to have an effect, which we discuss in the text (lines # 580-590). 

Discussion:

r617-619, 625 can be leaved out (see comments introduction). It is not relevant to compare is with the sex steroids in vertebrates.

***- As we explained in addressing similar suggestions for the Introduction, we believe that our paper is of interest to a broad readership, and therefore submitted it to a multidisciplinary biology journal rather than to a more specific entomological journal.

r668 The authors indicate “the first causal link between a specific gene and dominance in social insects”.  Is the gene directly involved in behaviours resulting in dominance (more discussion about dominance in queenless worker is needed including RHP)? What about the relation with ovarian activity and wax deposition. More discussion is needed as Kr-h1 is described as an early juvenile hormone-response gene. Are there suggestions for a pathway?  Is vitellogenin involved? The Kr-h1 down regulation in brain and fat body is not discussed (?)

***- These are very good questions for which our findings do not provide explicit answers. We assume that Kr-h1 is expressed in many tissues in which it mediates the influence of JH on the pattern of gene expression. We show that we successfully down-regulated kr-h1 expression in the fat body, ovaries (we added a new result panel showing this = supplementary fig 3C), and brain. Our results at the organismal level are not sufficient for determining if kr-h1 effect on dominance is specific to the brain or also requires interactions with other tissues such as the ovary or fat body. We also have no data on the effect of kr-h1 KD on the expression of other known or putative JH signaling genes. Thus, addressing the reviewer’s questions require us to be more speculative than we like to. Nevertheless, along with the reviewer comments, we elaborated a bit more in parts related to these questions (lines # 726-749).

The improved protocol is not discussed, as a large part of the paper concerns this topic.

***- We now refer to this contribution of our study in several parts of the Discussion. We also added a new paragraph summarizing the methodological contribution of our study (lines # 670-686).

Reviewer 2 Report

1. Summary: The manuscript presents a discussion on the universal phenomena of dominance rank for which molecular machinery is unknown. They used a eusocial insect - Bumblebee - to test whether a transcription factor related to a crucial growth hormone might be responsible for dominance rank in this species. The choice of this species is important since social insects such as bumblebee show a clear female hierarchal dominance in their colony by competing with non-queen female individuals. They found a growth hormone-related transcription factor mediates dominance behavior in bumblebee by knockdown technique. 
2. Strengths:
- First study linking dominance rank in social insects
- Tested the hypothesis in this insect species based on the effect of the analogous hormonal system in mammals on dominance rank
- Knockdown approach was adapted rather than mutagenesis for Kr-h1 to observe the effect on dominance rank which was important to minimize interference with developmental processes
- RNAi approach for bumblebees is fairly new (maybe started last 5-10 years) and in addition to that, this research used nano-particle-based RNAi protocol. So methodologically, this research is a successful example of deployment of the latest RNAi knockdown protocol which might help and motivate other insect researchers to employ this technique in their research

Weaknesses:
- Since the Kr-h1 transcription factor regulated multiple downstream pathways, this correlation might not be as straightforward as suggested
- They used Kr-h1 TF but there are other TFs that affect JH related downstream processes which were not considered in this research
- No other reference is available for other insects for Kr-h1 so it is difficult to verify the validity of this finding

3. Recommendations:
- Maybe add discussion of what additional variables might be involved in determining dominance rank in advanced eusocial or primitive social insect systems (compared to endocrine system of mammals that was focused on in the Introduction)
-No grammar/spell check suggestions

Author Response

  1. Summary: The manuscript presents a discussion on the universal phenomena of dominance rank for which molecular machinery is unknown. They used a eusocial insect - Bumblebee - to test whether a transcription factor related to a crucial growth hormone might be responsible for dominance rank in this species. The choice of this species is important since social insects such as bumblebee show a clear female hierarchal dominance in their colony by competing with non-queen female individuals. They found a growth hormone-related transcription factor mediates dominance behavior in bumblebee by knockdown technique.

  1. Strengths:

- First study linking dominance rank in social insects

- Tested the hypothesis in this insect species based on the effect of the analogous hormonal system in mammals on dominance rank

- Knockdown approach was adapted rather than mutagenesis for Kr-h1 to observe the effect on dominance rank which was important to minimize interference with developmental processes

- RNAi approach for bumblebees is fairly new (maybe started last 5-10 years) and in addition to that, this research used nano-particle-based RNAi protocol. So methodologically, this research is a successful example of deployment of the latest RNAi knockdown protocol which might help and motivate other insect researchers to employ this technique in their research

Weaknesses:

- Since the Kr-h1 transcription factor regulated multiple downstream pathways, this correlation might not be as straightforward as suggested

- They used Kr-h1 TF but there are other TFs that affect JH related downstream processes which were not considered in this research

- No other reference is available for other insects for Kr-h1 so it is difficult to verify the validity of this finding

***We actually cite quite a few studies in which Kr-h1 was knocked down in various insect species (lines # 155-162).

  1. Recommendations:

- Maybe add discussion of what additional variables might be involved in determining dominance rank in advanced eusocial or primitive social insect systems (compared to endocrine system of mammals that was focused on in the Introduction)

***- Added (lines # 55-61, 695-697, and 734-737)

-No grammar/spell check suggestions

Reviewer 3 Report

General comments

   In the work of Pandey and Bloch entitled “Krüppel-homologue 1 mediates hormonally-regulated dominance rank in a social insect”, the authors have studied the participation of the transcription factor Krüppel-homologue 1 (Kr-h1) in the regulation of the dominance behavior in the bumble bee Bombus terrestris using elegant approaches that include the combination of RNA interference and pharmacological techniques. The work deals with a very relevant issue since the precise participation of isolated genetic factors in complex behaviors such as dominance rank is not well understood in insects. The manuscript is well organized and written, and their objectives as well as the experimental design are clear and straightforward. The findings are compelling and the conclusions match the reach of findings. Some minor suggestions are included above.

Specific comments

- Page 2, line 95: The open parenthesis needs closing.

- Page 3, line 144: I believe that the authors meant to write “upregulated” instead of “unregulated”. Please check.

- Page 8, line 367: “B. terrestris” needs to be in italics.

- Page 13, lines 535-536: Perhaps the rationale of this experiment could be further explained, especially in comparison to experiments 1.

- Page 15, lines 601-602: The first sentence needs a citation.

- Supplementary figures: I found it odd the fact that, in the supplementary word file, the figures do not follow a logical order, i. e. supplementary figure 2 appears before supplementary figure 1. Please rearrange.

Author Response

General comments

   In the work of Pandey and Bloch entitled “Krüppel-homologue 1 mediates hormonally-regulated dominance rank in a social insect”, the authors have studied the participation of the transcription factor Krüppel-homologue 1 (Kr-h1) in the regulation of the dominance behavior in the bumble bee Bombus terrestris using elegant approaches that include the combination of RNA interference and pharmacological techniques. The work deals with a very relevant issue since the precise participation of isolated genetic factors in complex behaviors such as dominance rank is not well understood in insects. The manuscript is well organized and written, and their objectives as well as the experimental design are clear and straightforward. The findings are compelling and the conclusions match the reach of findings. Some minor suggestions are included above.

***- We are very pleased to learn about the reviewer positive assessment of our contribution.

 Specific comments

- Page 2, line 95: The open parenthesis needs closing.

***- Done (now line # 107).

- Page 3, line 144: I believe that the authors meant to write “upregulated” instead of “unregulated”. Please check.

***- Changed to upregulated, thanks! (line # 165).  

- Page 8, line 367: “B. terrestris” needs to be in italics.

***- Fixed (now line # 228).

- Page 13, lines 535-536: Perhaps the rationale of this experiment could be further explained, especially in comparison to experiments 1.

***- We thank the reviewer for this excellent suggestion which we addressed in lines 573-577.

- Page 15, lines 601-602: The first sentence needs a citation.

***- Citation included! (Now lines # 651-653).

- Supplementary figures: I found it odd the fact that, in the supplementary word file, the figures do not follow a logical order, i. e. supplementary figure 2 appears before supplementary figure 1. Please rearrange.

***- Corrected!
